# The Current State of Charcot–Marie–Tooth Disease Treatment

**DOI:** 10.3390/genes14071391

**Published:** 2023-07-01

**Authors:** Yuji Okamoto, Hiroshi Takashima

**Affiliations:** 1Department of Neurology and Geriatrics, Kagoshima University Graduate School of Medical and Dental Sciences, Kagoshima 890-8544, Japan; 2Department of Physical Therapy, School of Health Sciences, Faculty of Medicine, Kagoshima University, Kagoshima 890-8544, Japan

**Keywords:** Charcot–Marie–Tooth disease, PXT3003, gene therapy

## Abstract

Charcot–Marie–Tooth disease (CMT) and associated neuropathies are the most predominant genetically transmitted neuromuscular conditions; however, effective pharmacological treatments have not established. The extensive genetic heterogeneity of CMT, which impacts the peripheral nerves and causes lifelong disability, presents a significant barrier to the development of comprehensive treatments. An estimated 100 loci within the human genome are linked to various forms of CMT and its related inherited neuropathies. This review delves into prospective therapeutic strategies used for the most frequently encountered CMT variants, namely CMT1A, CMT1B, CMTX1, and CMT2A. Compounds such as PXT3003, which are being clinically and preclinically investigated, and a broad array of therapeutic agents and their corresponding mechanisms are discussed. Furthermore, the progress in established gene therapy techniques, including gene replacement via viral vectors, exon skipping using antisense oligonucleotides, splicing modification, and gene knockdown, are appraised. Each of these gene therapies has the potential for substantial advancements in future research.

## 1. Introduction

One of the major known genetically and heterogeneously transmitted peripheral neuropathies is the Charcot–Marie–Tooth (CMT) disorder, which affects individuals at any age. Patients typically exhibit distal muscle weakness and atrophy, weakened ankle dorsiflexion, lessened tendon reflexes, and noticeable foot arches (Pes cavus deformities). A slight to moderate distal sensory loss, mainly observed in a symmetric pattern known as the stocking–glove distribution, frequently coincides with muscle weakness [1]. CMT is a major hereditary neurological disorder, with an estimated incidence of 10 to 40 per 100,000 individuals. Based on this pattern, the neuropathies can be classified into three main types: autosomal dominant (demyelinating [CMT1] and axonal [CMT2]), X-linked (CMTX1), and autosomal recessive [2,3]. Nerve conduction velocity (NCV) can be enormously helpful in establishing a historical clinical segregation. A consistent slow NCV of < 38 m/s in the arms represents the demyelinating form of CMT1, whereas a value >38 m/s is distinctive of the axonal form of CMT2. Transitional NCVs (25–45 m/s) are often observed in males with CMTX1 and in patients with other transitional patterns of CMT [4].

The genetic diagnosis of CMT has been conducted by implementing the targeted next-generation sequencing and whole-exome sequencing approaches. Over 120 gene mutations are related to the pathogenesis of CMT and associated neuropathies [5,6]. The frequency of the various types of CMT varies by region and the age at which they were published. In the study by Saporta et al., 787 out of 1024 patients were diagnosed with CMT; a genetic subtype was identified in 527 (67%) of these patients, while the remaining 260 patients did not present with any identifiable mutation. Among the genetically defined cases, CMT1A (*PMP22*) emerged as the most prevalent subtype, comprising 55% of the cases, followed by CMTX1 (*GJB1*; 15.2%), hereditary neuropathy with liability to pressure palsy (HNPP; 9.2%), CMT1B (*myelin protein zero* (*MPZ*); 8.5%), and CMT2A (*MFN2*; 4%) [7]. DiVincenzo et al. described the frequency, detection rate, and mutation types in 14 representative genes (*PMP22*, *GJB1*, *MPZ*, *MFN2*, *SH3TC2*, *GDAP1*, *NEFL*, *LITAF*, *GARS*, *HSPB1*, *FIG4*, *EGR2*, *PRX*, and *RAB7A*) related to CMT in a cohort study of 17,880 patients tested in a commercial genetic laboratory. Genetic anomalies were detected in 18.5% (n = 3312) of the total population. Sanger sequencing and multiplex ligation-dependent probe amplification revealed that duplications (56.7%) or deletions (21.9%) in *PMP22* were reported for most of the positive results; next, mutations were detected in *GJB1* (6.7%), *MPZ* (5.3%), and *MFN2* (4.3%). *GJB1* deletions and mutations in the residual genes were 5.3% of the anomalies. Of the individuals presenting a positive genetic outcome in a CMT-related gene, 94.9% had a mutation in one of the following four genes (*PMP22*, *GJB1*, *MPZ*, or *MFN2*) [8].

Effective pharmacological treatments for CMT are currently lacking. This review discusses the treatment of CMT with the four main causative genes. Numerous factors, such as myelin development and conservation, transcription factors for myelin genes, gap junctions and channels, axonal transport (both retrograde and anterograde, involving kinesins, dynein, and dynactin), mitochondrial dynamics, and vesicle trafficking contribute to the pathomechanisms of CMT and associated neuropathies. Therapeutic strategies may target specific defects in select CMT subtypes or focus on addressing shared pathomechanisms via a broader approach applicable to various categories of CMT and other neuropathies [9]. The challenges in developing an effective treatment for CMT can be attributed to three main factors: (1) extensive genetic heterogeneity, with more than 1500 identified underlying point alterations, counting the 1.4 Mb CMT1A duplication, coupled with overlapping disease phenotypes; (2) the relative scarcity of individuals per genotype, which reduces interest from both research communities and pharmaceutical companies; and (3) the complexity involved in transitioning preclinical research from rodent and cellular models to human clinical investigations [5]. This study explores the current landscape of potential therapeutic interventions for CMT, encompassing agents under clinical evaluation that show promise for future clinical use and notable compounds in the preclinical pipeline expected to catalyze the continued research efforts. Thus, building on the momentum of the past two decades, significant strides have been made toward developing gene therapies for CMT. Preclinical studies have established a proof of concept for specific therapeutic approaches, underpinning their potential for clinical translation. This has been further bolstered by recent successes in gene therapy for other neuromuscular disorders, as evidenced by clinical trials for TTR-related hereditary amyloidosis and spinal muscular atrophy (SMA). These advancements have invariably invigorated the pursuit of gene therapy for CMT neuropathies.

## 2. The Different Forms of Major CMTs

### 2.1. Demyelinated CMT

#### 2.1.1. CMT1A (PMP22)

Peripheral myelin protein-22 (PMP22), a 22-kDa intrinsic tetraspan glycoprotein, is mainly observed in the compact myelin of the peripheral nervous system. PMP22 is principally produced for myelin creation and preservation by myelinating Schwann cells (SCs) during growth and accounts for about 2%–5% of the compact myelin in the peripheral nervous system. This protein is essential for myelogenesis, myelin thickness, the growth and differentiation of SCs, and maintaining the axons and myelin of the PNS [1]. It is particularly relevant in the context of inherited peripheral neuropathies, comprising >50% of the total cases, comprising CMT1A, hereditary neuropathy with liability to pressure palsy (HNPP), and CMT1E. Alterations in PMP22 levels due to gene mutations, such as trisomy in CMT1A, heterozygous deletion in HNPP, and point mutations in CMT1E, lead to varying phenotypes; overexpression and point mutations result in gain-of-function effects and deletion produces loss-of-function. CMT1A, the major known form of CMT, is the result of a 1.4 Mb *PMP22* duplication on chromosome 17p11.2, leading to disrupted myelin formation and compromised nerve function. Patients typically present a “classical CMT phenotype” within the first 20 years of life, characterized by progressive muscle feebleness, atrophy, decreased sensory function, hyporeflexia, and skeletal malformations. Although the disorder progresses slowly and patients may require ankle–foot orthotics, they generally maintain ambulation and experience no significant impact during their life span. Extensive research on *PMP22* and its role in CMT1A has advanced our knowledge of the illness and improved the potential therapeutic techniques [3].

#### 2.1.2. CMT1B (MPZ)

CMT1B accounts for 10% of the total CMT1 cases and arises from *MPZ* mutations, located on chromosome 1q22-q23; the majority of the peripheral nerve myelin protein is encoded by CMT1B. The role of Po protein as a homophilic adhesion molecule is to enable myelin compaction [10]. *MPZ* variants lead to the etiology of demyelinating neuropathy CMT1B (OMIM 118200). Some mutations result in axonal neuropathy CMT2I/J (OMIM 607677/607736) and the additionally critical juvenile-onset Dejerine–Sottas syndrome (OMIM 145900) and congenital hypomyelination neuropathy 2 (OMIM 618184). Furthermore, *MPZ* variants are related to the dominant intermediate CMT disease D (CMTDID; OMIM 607791). The phenotype of CMT resulting by *MPZ* variants varies from major pediatric-onset to minor adult onset. Approximately 250 variants of this gene have been identified as causes of inherited peripheral neuropathy [11].

#### 2.1.3. CMTX1 (GJB)

The most known X-linked form of CMT disease and the second most common form of CMT is the X-linked CMT disease type 1 (CMTX1), caused by mutations in *GJB1*, which encode gap junction protein β 1, also recognized as connexin 32 (Cx32). The latter is expressed by SCs and oligodendrocytes, whereas the gap junction formed by Cx32 is critical in the homeostasis of myelinated axons [12]. While men present moderate to severe symptoms, heterozygous women are generally less affected. CMTX1 is speculated to be an X-linked dominant trait because it affects female carriers. In general, females are affected with a minor version of the phenotype, and the onset is delayed compared to males. Clinically, CMTX1 may be largely represented as demyelinating or axonal neuropathy, although previous electrophysiological and pathological researches propose a more prominent axonal endorsement. Further to peripheral neuropathy, severe episodic central nervous system dysfunctions are distinctive of CMTX1 and usually manifest with numerous combinations of paralysis, dysarthria, dysphagia, ataxia, dyspnea, somnolence, and behavior anomalies [3,13].

### 2.2. Axonal CMT

#### CMT2A (MFN2)

The location of the mitofusin 2 gene (MFN2) mutations are on chromosome 1p36 and are accountable for CMT2A, the most dominant form of CMT2 (approximately 20% of the total cases of CMT) [3]. A previous study revealed that CMT2A constituted the majority (91%) of strictly affected CMT2 patients and showed a smaller proportion (11%) of the mild to moderate cases [14]. The inherited form is mainly autosomal dominant (AD) [15], with sporadic autosomal recessive or semidominant cases caused by both homozygous and compound heterozygous *MFN2* mutations [16]. *MFN2,* a GTPase protein fixed in the outer membrane of mitochondria, has a mediated function by the two transmembrane domains located near the C-terminus [17]. Mutations in *MFN2* result in irregular mitochondrial aggregation and occupation accompanied by dysfunctional subcellular mitochondrial trafficking. In CMT2A patients, the most affected nerves are the peripherals comprising longer axon projections, probably because of advanced energy demands compared to other cell types.

## 3. CMT Treatment with Compounds and Drugs (Table 1)

### 3.1. Clinical Research (Previous and Current)

#### 3.1.1. Ascorbic Acid

Ascorbic acid (AA; Vitamin C) was among the initial therapies explored for CMT1A. AA therapy has been used on a large scale in several countries. Although the effectiveness of this therapy has not yet been verified, it has contributed valuable insights into the treatment strategies for CMT. It is crucial to note that safety and acceptability also played a role in the widespread adoption of AA therapy. Many intractable neurological disorders require a longer duration to exhibit the efficacy of a treatment. Preclinical research studies have emphasized the significance of PMP22 dosage in the CMT1A pathogenesis; therefore, therapeutic efforts have focused on decreasing this gene’s expression and endorsing effective myelination [17]. AA is recognized for its antioxidant properties, neuromodulatory actions, and critical function in myelination. Increased concentrations of AA block adenylate cyclase activity, the enzyme responsible for making cyclic adenosine monophosphate [18]. In vivo studies in C22 mice verified that the expression of PMP22 is suppressed by AA and confirmed the motor function improvement in treated animals [19]. These encouraging findings, coupled with the well-established safety profile of AA, led to multiple clinical trials. Several human studies have been conducted in CMT1A patients, testing various doses of vitamin C (1 to 4 g/day) for up to two years in both adults and children; however, these trials did not indicate the clinical efficacy of AA in patients with CMT1A [20].

**Table 1 genes-14-01391-t001:** Outline of the treatment methods for Charcot–Marie–Tooth disease.

Compound	CMT Type	Mechanism	Clinical Trials
Ascorbic acid	CMT1A	Reduces PMP22 expression via the inhibition of cAMP pathway	Phase III studies concluded; did not show a significant effect
PXT3003 (a combination of low doses of baclofen, sorbitol and naltrexone)	CMT1A	Inhibits the proliferation of Schwann cells and downregulates the synthesis of PMP22; baclofen, and GABAB receptor modulator	Phase III unpublished. New Phase III requested by FDA
Progesterone antagonists (onapristone, ulapristal)	CMT1A	Inhibits myelin-related genes expression in SCs	Onapristone: unacceptable side effects. Ulapristal: phase II trial conducted
P2X7 receptor modulators (A438079)	CMT1A	Reduces excessive calcium influx into Schwann cells	P2X7 antagonist acceptable safety and tolerability in a previous phase II trial in rheumatoid arthritis
Dietary lipid supplementation	CMT1A	Diet-based correction of defective myelin lipid biosynthesis	Trial with oral lecithin supplementation planned
ACE083	CMT1, CMTX1	Myostatin pathway	Phase I+II trial did not produce significant clinical improvement
Curcumin (Nano-Cur), sephin-1 (IFB-088)	CMT1A, CMT1E, CMT1B	Rescue ER accumultion misfold protein by UPR activation	N.A
Melatonin	CMT1A	Alleviates hyperoxidative and inflammatory conditions	small pilot study was conducted
Fasting and rapamycin	CMT1A	Improve ER processing	N.A
Neuregulin pathways (Neuregulin-I III)	CMT1A, CMT1B, CMT4B, HNPP	Regulates thickness of myelin	Niacin-niaspan candidate
Sodium channel blockers	CMT1B	Blocking of Nav 1.8 channel	Lamotrigine could be a candidate compound
HDAC6 inhibitors (CKD504)	CMT2F, dHMN2	Reduces microtubles acetylation, action axonal transport	N.A
SARM1 inhibitors	CMT2	Prevents axonal degeneration	N.A
CSF1R inhibitors, CSF1 receptor antagonists	CMT1A, CMT1B, CMTX1	Decline in nerve macrophages	N.A
eIF2α and Gadd34	CMT1B	Upregulation of eIF2α phosphorylation controlling translation	N.A
Sox2 and Id2	CMT1B	Negatively regulates myelination,	N.A
Isoquinoline	CMT2A	Inhibits the activity of the SARM1 NADase	N.A
MFN2 agonists	CMT2A	Improves mitochondrial trafficking	N.A
6-phenylhexanamide derivative mitofusin activators	CMT2A	Improves mitochondrial motility	N.A

cAMP, cyclic adenosine monophosphate; CMT, Charcot–Marie–Tooth disease. CSF1R, colony-stimulating factor 1 receptor; eIF2α, eukaryotic translation initiation factor 2α kinase; ER, endoplasmic reticulum; FDA, Food and Drug Administration; GABAB, gammaaminobutyric acid B receptor; GADD34, growth arrest and DNA damage gene; HDAC6, histone deacetylase 6; dHMN, distal hereditary motor neuropathy; HNPP, hereditary neuropathy with liability to pressure palsies; Id2, Inhibitor of DNA binding 2; MFN2, mitofusin 2; N.A., not applicable; NAD, nicotinamide adenine dinucleotide; SARM1, sterile α and TIR motif containing 1; SCs, Schwann cells; UPR, unfolded protein response. This table was adapted with some changes from C. Pisciotta et al., *Expert Rev. Neurother.*
**2021**, *21*, 701–713 [21] and M. Stavrou et al., *Int. J. Mol. Sci.*
**2021**, *22*, 6048 [1].

#### 3.1.2. PXT3000

PXT3003 is a new fixed dose synergistic mixture of baclofen, naltrexone, and sorbitol, expressed as an oral solution administered two times daily. The selection of drugs for CMT1A polytherapy was made from a group of approved medications for unassociated diseases by means of a systems biology method followed by pharmacological safety considerations. PXT3003 has been empirically shown to attenuate the levels of Pmp22 mRNA, facilitate the myelination process, enhance the balance in PI3K-AKT/MEK-ERK signaling pathways, augment functional neuromuscular junction numbers, and foster the differentiation of Schwann cells. A phase III trial expressed worries regarding the steadiness of a high concentration of PXT3003, leading to its termination (NCT02579759). Subsequently, a novel clinical test was initiated in 2021 (NCT04762758), in which a determined concentration of PXT3003 was orally given two times daily. High-dose PXT3003 significantly improved the Overall Neuropathy Limitations Scale total score compared to placebos [1,22,23,24].

#### 3.1.3. Progesterone Receptor Antagonist

P_0_ promoter and promoter 1 of the *PMP22* gene in SCs upregulates the expression of *SOX-10* and *KROX-20*, further driving PMP22 expression. Furthermore, progesterone derivatives induce myelin gene expression by γ-aminobutyric acid A receptor (GABA_A_) receptors activation in SCs [25] Onapristone, classified as a progesterone receptor adversary, led to a reduction in PMP22 and amplified axonal configurations, consequentially mitigating behavioral irregularities within a CMT1A rat model [26]. Clinical experiments of Onapristone have not been applied in CMT1A patients because of the critical disadvantages detected in cancer patients treated with this medicine. In France, a Phase II trial of Ulapristal was undertaken but fell short in achieving the intended patient count, and the results remain undisclosed (NCT02600286) [1,22].

#### 3.1.4. ACE-083

Acceleron Pharma spearheaded an innovative therapeutic approach for focal or asymmetric myopathies by developing ACE-083, a locally functioning, follistatin-based fusion protein that adopts muscle growth and functionality by sequestering specific ligands extracellularly. Preclinical studies have shown that ACE-083 triggered localized, dose-responsive hypertrophy in the injected muscle, devoid of any systemic muscular effects or endocrine interference. This was observable in both wild-type mice and mouse models of CMT. Additionally, ACE-083 amplified the force of isometric contractions in the anterior tibialis muscle and augmented ankle dorsiflexion torque in CMT mice, suggesting its potential efficacy in treating CMT patients. A Phase I dose escalation trial was subsequently conducted, followed by a Phase II study, where up to 250 mg of ACE-083 was administered bilaterally into the anterior tibialis muscles every three weeks for a maximum of nine doses. The study comprised 62 patients diagnosed with CMT1 and CMTX1. Unfortunately, the program was prematurely terminated before advancing to phase III due to disappointing results; despite an observed increase in muscle volume, this physiological change failed to yield significant enhancements in practical or quality-of-life procedures when related with the placebo (NCT03124459) [27].

#### 3.1.5. P2X7 Purinoreceptors

Intracellular Ca (2+) concentration within SCs from CMT1A rats demonstrated abnormal elevation, an effect attributed to *Pmp22*-induced overexpression of the P2X7 purinoceptor. A pharmacological P2X7 receptor antagonist, A438079, fostered improved myelination in dorsal root ganglion cultures from CMT1A rats in vitro. The administration of A438079 resulted in a notable enhancement of muscle strength in CMT1A-affected rats, in significant contrast to placebo-controlled subjects. Notably, a marked upsurge in the total count of myelinated axons in the tibial nerves was unveiled through histological examination. A438079 appeared to remedy the defect in SC differentiation observed in CMT1A rats. The results of this study propose that the pharmacological suppression of the P2X7 receptor is tolerable in CMT1A rats, thus offering an evidence-based suggestion that inhibiting this pathway could rectify the molecular abnormalities and enhance the clinical presentation of CMT1A neuropathy [28]. The transition to clinical experiments for CMT1A is made smoother by prior Phase II clinical trials involving rheumatoid arthritis, where a P2X7 antagonist demonstrated satisfactory safety and tolerability [29].

#### 3.1.6. Lipid Supplementation

In a rat model of CMT1A, myelinating SCs display a developmental deficiency characterized by lessened genes transcription necessary for myelin lipid biosynthesis. The supplementation of phosphatidylcholine and phosphatidylethanolamine in the diet of rats enhanced biosynthesis of myelin and nerve function. While clinical experiments have shown no disadvantages from this therapy, it is still uncertain if high amounts of dietary phospholipids can help CMT1A patients [21]. Clinical translation is feasible due to the absence of any significant side effects with dietary phospholipids; therefore, a clinical experiment with oral lecithin supplementation has been set [22].

### 3.2. Preclinical Research

#### 3.2.1. Neuregulin-1 Type I (NRG1)

The peripheral myelination of the nerves is endorsed by axonal neuregulin 1 type III (Nrg1-III) through downstream signaling pathways activation, such as PI3K/Akt and mitogen-activated protein kinase (MAPK)/ERK, which ultimately regulate the master transcriptional regulators of myelin genes, such as *Krox20* [30]. The thickness of the myelin is controlled by the amount of Nrg1-III, whose action is moderated by the secretases β-site amyloid precursor protein cleaving enzyme 1 (BACE1) and tumor necrosis factor-α converting enzyme (TACE) through distinct extracellular cleavage mechanisms [9]. Niacin/Niaspan (nicotinic acid) may improve focal hypermyelination, avoid myelin deterioration, and reserve axonal physiology by increasing TACE activity and downregulating Nrg1 type III. Niacin/Niaspan has been shown to reduce hypermyelination in the vimentin-null model and is related to improved Nrg1 type III expression. Additionally, Niaspan was reported to reduce myelin out folding in the nerves of *Mtmr2*−/− mice, a model of CMT4B1 neuropathy, and diminish tomacula in the nerves of *Pmp22*+/− mice, a model of HNPP neuropathy [31]. In a mouse model of demyelinating CMT1B, the neurophysiological and morphological parameters were promoted by the genetic overexpression of Nrg1-III without aggravating the toxic gain-of-function underlying the neuropathy. Moreover, stimulation of Nrg1-III signaling through pharmacological repression of the Nrg1-III inhibitor TACE enhanced neuropathy [30]. Ameliorated PI3K-Akt signaling by axonally overexpressed Nrg1 leads to contaminated SCs to differentiate and conserves peripheral nerve axons. Notably, in a preclinical treatment trial using a CMT1A rat model, soluble NRG1 successfully overcame impaired peripheral nerve development and restored axon survival into adulthood when treatment was limited to early postnatal development [32]. Thus, modulating Nrg1-III levels is a potential method for treating both hypomyelinating and hypermyelinating neuropathies.

#### 3.2.2. Curcumin

Mutant protein retention in the endoplasmic reticulum (ER) triggers the unfolded protein response (UPR) activation, an adaptive and protective mechanism to alleviate stress caused by misfolded proteins. Curcumin, a polyphenolic compound found in the dietary spice turmeric, exhibits a range of pharmacological effects, including anti-inflammatory, antioxidant, antiproliferative, and antiangiogenic properties. Despite its low bioavailability, the effective treatment of curcumin against several human disorders such as cancer, cardiovascular diseases, and diabetes has been documented [33]. Many myelin gene mutations causing severe diseases, including those in *MPZ* and *PMP22*, produce aberrant proteins that predominantly accumulate in the ER, leading to SC apoptosis and, subsequently, peripheral neuropathy. In an in vitro study, Khajavi et al. demonstrated that supplementation with curcumin could counteract ER retention and aggregation-induced apoptosis related to *MPZ* and *PMP22* mutants. Moreover, they showed that oral curcumin administration partially alleviated the severe neuropathy phenotype of the Trembler-J (Tr-J) mouse model (with the L16P mutation, a model of CMT1E) in a dose-dependent manner, featuring dominantly transmitted *Pmp22* missense mutations [34]. Okamoto et al. observed the activation of different UPR branches in Tr-J mice and found that curcumin therapy yielding a reduction in the expression of the UPR marker, suggesting that it mitigated ER stress in the sciatic nerves of the mice [35]. Additionally, curcumin melted in sesame oil or phosphatidylcholine ameliorated peripheral neuropathy in R98C mice, an accurate model of CMT1B, by lessening ER stress and UPR activation, while promoting SC differentiation [36]. Curcumin–cyclodextrin/cellulose nanoparticles (Nano-Cur) were developed to overcome the limited pharmacokinetics of curcumin [37]. The effect of Nano-Cur was test in vitro and in vivo in a rat model of CMT1A and it was found that the level reactive oxygen species declined. Moreover, improvements in mitochondrial membrane potential and integrity were observed, resulting in improved myelination and nerve function [38].

#### 3.2.3. Sephin-1

Sephin-1 (IFB-088), which inhibits eIF2a dephosphorylation (a kinase in the PERK arm of the UPR) and lengthens protein translation reduction in response to stress, showed to be beneficial in both S63del and R98C *Mpz* mouse models by sustaining the response and avoiding molecular, morphological, and motor flaws of neuropathy [39].

At present, Sephin-1 is being evaluated in CMT1A rodent models, making it a potential candidate for clinical experiments in both CMT1A and CMT1B [22]. Theoretically, this technique may also be useful for other mutants retained in the ER.

#### 3.2.4. Eukaryotic Initiation Factor 2-Phosphorylation (eIF2α) and Gadd34

Human CMT1B neuropathy and similar demyelinating in transgenic mice are caused by the mutant *P0*S63del, which is reserved in the ER, activating a UPR. In the presence of ER stress, protein kinase R-like endoplasmic reticulum kinase (PERK) phosphorylates eIF2α to decrease the global translation, thus lessening the misfolded protein overload in the ER. Genetic and pharmacological inactivation of Gadd34, a subunit of the PP1 phosphatase complex that enables the dephosphorylation of eIF2α, was reported to extend eIF2α phosphorylation and lessen motor, neurophysiological, and morphological deficits in S63del mice [40]. Consequently, silencing Gadd34 or directly increasing eIF2 phosphorylation may be useful on CMT1B.

#### 3.2.5. Melatonin

Melatonin possesses properties of an antioxidant and anti-inflammatory agent. Supplements containing melatonin have received Food and Drug Administration (FDA) approval and have been widely utilized in clinical investigations. In a trivial pilot study including three CMT1A patients treated with melatonin, plasma levels of lipid peroxidation (LPO), nitrites (NOx), and IL-1β, IL-2, IL-6, TNF-α, INF-γ, the ratio of oxidized to reduced glutathione (GSSG/GSH), and the activities of superoxide dismutase (SOD), glutathione-S-transferase (GST), glutathione peroxidase (GPx), and reductase (GRd) in erythrocytes were evaluated. The outcomes presented amplified SOD, GST, GPx, and GRd activities in CMT1A patients, which subsequently decreased after 3 and 6 months of treatment. The GSSG/GSH ratio increased significantly in the patients, but returned to normal levels after melatonin treatment. The presence of inflammation was verified by the elevated levels of all the proinflammatory cytokines measured, which were regulated by melatonin; in addition, elevated LPO and NOx levels in the patients were also regulated by melatonin. These findings indicate that melatonin may have therapeutic effects on CMT1A patients by alleviating hyperoxidative and inflammatory conditions and reducing the degenerative process [1,41].

#### 3.2.6. HDAC6 Inhibitor

The microtubules acetylation is imperative for axonal transport. Defective axonal transport is observed in numerous neuropathy models, comprising mice with mutations in the *HSPB1* gene. *HSPB1* encodes heat shock protein (HSP27), and mutations in this gene have been implicated in axonal CMT disease type 2F and distal hereditary motor neuropathy (dHMN). HDAC6 inhibitor has been displayed to correct axonal transport defects and save the phenotype of *HSPB1* mutant mice [42]. Furthermore, the HDAC6 inhibitor controls the acetylation of nuclear and cytosolic proteins [43], comprising HSP90 and HSP70, which contribute to the process of protein conformation, including that of PMP22 [44,45]. The deletion of *Hdac6* prohibited motor and sensory dysfunctions in an MFN2 mouse model of CMT2A. These outcomes collectively propose that reduced acetylated α-tubulin could characterize a shared pathomechanism among various axonal neuropathies, with HDAC6 inhibitors potentially offering therapeutic benefits for these conditions [46]. HDAC6 is implicated in various conditions, including tumors, neurological disorders, and inflammatory diseases; consequently, targeting HDAC6 has become a potential therapeutic approach recently. ACY-1215 (ricolinostat) is the first orally existing extremely selective HDAC6 inhibitor, and its efficiency and beneficial effects are being continuously confirmed [47].

Recessive CMT phenotypes are mainly generated when gene function declines or is lost. Gene replacement treatment may enhance effectiveness once the timing and cell delivery issues are determined. However, dominant CMT phenotypes often have a mutant allele toxic to the peripheral nerve; therefore, gene knockdown may be mandatory to improve the phenotypes [48].

#### 3.2.7. Fasting and Rapamycin

The Tr-J mouse model of CMT1A was put on an intermittent fasting regimen, which improved their locomotor performance compared to the control group [49]. The functional benefits of dietary restriction include augmentation of the myelin proteins expression, increased thickness of myelin sheath, reduced redundancy in basal lamina, and downregulated aberrant SC proliferation. These findings show that dietary limitation is helpful for peripheral nerve function in Tr-J neuropathic mice since it encourages the conservation of locomotor performance [49]. However, intermittent fasting is not a clinically available treatment; therefore, studies using rapamycin, an FDA-approved calorie restriction mimetic, were conducted. The drug was administered to explant cultures of C22 mice, which improved processing of PMP22 and enhanced myelin internodal profile; it also increased the production of other myelin-related proteins; nevertheless, despite these potential in vitro outcomes, rapamycin did not recover neuromuscular performance in the Tr-J mouse model in vivo [50].

#### 3.2.8. Sox2 and Id2

The myelination status of SCs appears to be detected by the stability between opposing signaling systems. Positive regulators such as Krox-20, Oct-6, Sox-10, Brn2, and NF-κB are predominant in normal nerves, but the balance shifts to negative regulators such as c-Jun, Notch, Pax-3, Sox2, and Id2 in the injured and pathological nerves. In addition, negative regulators may be important in the onset and myelination rate during growth (59). The continued expression of Sox2 and Id2 seems to perform a protective function in neuropathy since their elimination in *P0*S63del mice—a CMT1B mouse model—led to an exacerbation of the dysmyelinating phenotype [51]. Therefore, the overexpression of these genes may have a therapeutic effect on CMT1B.

#### 3.2.9. Colony-Stimulating Factor 1 Receptor Inhibitor

The neuropathic phenotype is promoted by low-grade inflammation from phagocytizing macrophages in mice models for CMT1A, CMT1B, and CMTX1. Colony-stimulating factor (CSF1) is a macrophage activator expressed by endoneurial fibroblasts. It mediates macrophage-related neural impairment in mice and humans with CMT. Consequently, oral administration of the CSF1 receptor (CSF1R) inhibitor PLX5622 followed by a vigorous decrease in the number of macrophages in the peripheral nerves of Cx32def mice; moreover, long-term CSF1R inhibition was reported to improve axonal integrity in these mice. However, although long-term CSF1R inhibition reduced the neuropathic patterns in *P0*het mutants, it did not produce the same effect in *PMP22*tg mutants [52].

#### 3.2.10. Sodium Channel Blockers

Dysmyelination in *Mpz* knockout mice associates with the ectopic expression of the sensory neuron-specific sodium channel isoform NaV1.8 on motor axons. Progressive impairment of motor performance in MPZ-deficient mice can be inverted by NaV1.8 blocker treatment [53]. Oral subtype-selective NaV1.8 blockade can treat severe demyelinating motor dysfunction, including CMT1B and probably other demyelinating CMT types [54].

#### 3.2.11. SARM1 Pathway

Axonal degeneration, a shared endpoint across all CMT types, occurs irrespective of the primary pathology (i.e., myelinopathy with secondary axonal damage or primary axonopathy). Axons are hardwired to self-destruct under specific circumstances, such as stress, disease, or during the developmental phases. Gerdts et al. proposed a biochemical mechanism that regulates axonal degeneration by manipulating sterile Aapha and TIR Mmtif Containing 1 (SARM1) variants, which could be alternately activated or inhibited within cells [55]. Axonal degeneration, a defining feature of numerous neurological disorders, including CMT2A, is conceptualized as a hereditarily encoded program of subcellular self-destruction in which the SARM1 protein has an essential function. Activation of SARM1 instigates axonal degeneration, even without any discernible damage [56]. A reduction in mitochondrial membrane likely precipitates a lower survival factor in the axon nicotinamide mononucleotide adenylyl-transferase 2, which triggers SARM1 and leads to axonal degeneration [57]. Inhibitors of SARM1, currently under development, hold significant potential for treating all CMT types and related neuropathies. A gene-based therapeutic approach utilizing dominant-negative SARM1 mutants encapsulated synergistically within an AAV8 capsid has been recently introduced [58]. Postmitochondrial injury, axonal rescue, and recovery are feasible with rotenone via SARM1 inhibition, even when axons have already transitioned into a metastable state [59]. Consequently, SARM1 inhibition continues to be a primary therapeutic target for CMT2A and other types of axonal CMT and demyelinating forms that cause the degeneration of secondary axons [1].

#### 3.2.12. MFN2 Agonist and MFN1

Crucially, MFN2 shares significant homology with MFN1; research across diverse experimental systems has revealed their direct interaction capabilities and cooperative role in mitochondrial fusion, in addition to the potential of MFN1 to reimburse for an MFN2 shortage. The low expression of MFN1 in axons elucidated the heightened susceptibility of motor and sensory neurons to MFN2 mutations [60]. The dimerization of mitofusin molecules on neighboring mitochondria is enabled by the transition of the HR2 domain from a “closed” state, where it binds to HR1 of the same MFN2 molecule, to an “open” state, where it binds to HR2 of a different MFN2 molecule. In a recent study, molecules that target HR1/HR2 of MFN2, dubbed “mitofusin agonists”, could shift the mitofusin equilibrium toward the active/open state and improve mitochondrial axon trafficking deficits in cultured motor neurons from Mfn2 T105M mutant mice [61]. MiM111, an orally administered mitofusin agonist, alleviated the disease symptoms in Mfn2 T105M mutant mice [62]. Given the pervasive negative influence of mutant MFN2 when the quantity of the MFN1 isoform is diminished (as seen in neurons), elevating the concentrations of either MFN1 or WT MFN2 could potentially act as a beneficial therapeutic strategy for CMT2A [63]. Remarkably, the nervous system of Mfn2 R94Q mutant mice responded to MFN1 overexpression with an increase in body weight, enhanced behavioral responsiveness and visual sharpness, prolonged survival, and a reduction in mitochondrial clustering and axonal degeneration. Hence, it is evident that the MFN1/MFN2 ratio is a critical determinant of neuronal vulnerability to the detrimental effects of mutant MFN2. Augmenting the levels of MFN1 or WT MFN2 could be a feasible beneficial approach for CMT2A [63].

### 3.3. CMT Treatment with Gene-Mediated Therapy (Table 2)

#### 3.3.1. Viral Vector-Based Therapy and the Growth Factors Neurotrophin 3

Gene therapy involves the genetic material delivery to an individual, predominantly via viral vectors. One significant advantage of viral vector genetic treatment is its ability to cross both the blood–brain barrier and the blood–nerve barrier, in addition to being a one-time therapy that offers long-term effects. Lentiviral vectors were among the first to enable the efficient delivery of therapeutic genes for CMT models treatment. Clinical experiments have shown the local use of integrative lentiviral vectors for other illnesses; however, their applicability for CMT-related neurological conditions is still incomplete due to their integration into the host genome [64]. Given that both neurons and SCs targeted by CMT therapies are extremely differentiated and non-proliferating, the episomal persistence of adeno-associated viruses (AAVs) without integration into the host genome does not compromise the treatment stability. AAV vectors have been used in promising preclinical study to treat CMT neuropathies, either by bringing the expression of trophic factors or by targeting the culpable genes in neurons or SCs. AAV1 has been employed for a CMT1A clinical experiment (NCT03520751), while other serotypes such as AAV9 have been clinically applied to treat other neuromuscular diseases, particularly SMA (NCT03306277) [65].

**Table 2 genes-14-01391-t002:** Outline of the gene-mediated treatment methods for Charcot–Marie–Tooth disease and associated neuropathies.

Compound	CMT Type	Mechanism	Clinical Trials
genomic HGF cDNA hybrid (VM202)	CMT1A	Stimulates SC repair and regeneration	Phase I+II studies concluded (NCT05361031)
AAV1 delivered NT3	CMT1A, CMT1X	Expresses neurotrophic factor	Phase I+II studies concluded (NCT03520751)
siRNA	CMT1A	Allele specific downregulation of the overexpression of PMP22	N.A.
siRNA (P2RX7)	CMT1A	Reduces abnormal Ca^2+^ influx into SC	N.A.
siRNA conjugated to squalenoyl nanoparticles	CMT1A	Downregulates PMP22 overexpression	N.A.
shRNA	CMT1A	Downregulates PMP22 overexpression	N.A.
Lentiviral delivered miR-318	CMT1A	Overexpression of miR-318 downregulates overexpressed PMP22	N.A.
AAV2 delivered miR-29a	CMT1A	Overexpression of miR-29a downregulates overexpressed PMP22	N.A.
ASOs	CMT1A	Downregulates PMP22 overexpression	N.A.
Antiparallel triplex-forming oligonucleotides	CMT1A	Bind on PMP22 promoters to downregulate overexpressed PMP22	N.A.
CRISPR/Cas9	CMT1A	Deletes TATA-box of PMP22 gene promoter to downregulate PMP22 overexpression	N.A.
Lentiviral delivered GJB1	CMT1X	Schwann cell specific Cx32 production	N.A.
AAV9 delivered GJB1	CMT1X	Induces Schwann cell-specific Cx32 production	N.A.
AAV8 delivered SARM1 mutants	CMT2A	Block the wild type SARM1 function	N.A.
MFN1 genetic addition	CMT2A	Compensates the dysfuction of mutated MFN2	N.A.

AAV, adeno-associated virus; ASO, antisense oligonucleotide; cDNA, complementary DNA; CRISPR/Cas9, clustered regularly interspaces short palindromic repeats/CRISPR-associated protein 9; HGF, Hepatocyte Growth Factor; MFN1, mitofusin1; miRNA, microRNA; N.A, not applicable; NT-3, neurotrophin-3; SARM1, sterile α and TIR motif containing 1; SC, Schwann cell; shRNA, short hairpin RNA; siRNA, small interfering RNA. This table was adapted with some changes from C. Pisciotta et al., *Expert Rev. Neurother.*
**2021**, *21*, 701–713 [21] and M. Stavrou et al., *Int. J. Mol. Sci.*
**2021**, *22*, 6048 [1].

The growth factor neurotrophin 3 (NT-3) is most frequently used in therapeutic studies with viral vectors; it promotes nerve regeneration following injury and SC survival. Consequently, the injection of recombinant NT-3 has been found to enhance regeneration and remyelination in animal models [65]. SCs expressing endogenous NT-3 contribute the autocrine loop, permitting the cells to develop and endure without the axon and motivating neurite outgrowth and myelination [66,67,68,69]. A pilot study involving eight patients with CMT1A demonstrated increased myelinated fiber density, reduced neurological disability, and improved sensory scores compared to a placebo [70]. Nevertheless, long-term therapy was not possible due to its highly short serum half-life. Additional investigations have furnished preclinical data indicating the effectiveness of AAV-mediated NT-3 genetic treatment (AAV1-NT-3) in a mouse model of CMT1A and more lately in Cx32 knockout mice [65,71,72]. Therefore, a study has been devised to intramuscularly administer rising dosages of the AAV1-NT-3 gene in both legs of CMT1A subjects (NCT03520751). Gene therapy aimed at replacing the defective gene is under investigation for various types of CMT characterized by loss-of-function mutations, such as CMTX1 and the recessive forms of CMT [22]. Intrathecal delivery of GJB1 was successfully carried out by Kleopa et al. using either lentivirus or AAV9 vectors along with the myelin-specific *Mpz* promoter in *GJB1*-knockout mice lacking Cx32 expression. This gene delivery yielded a stable Cx32 expression in SCs and peripheral nerves, accompanied by clinical improvement. However, owing to the possibility of damaging interactions between the delivered wild-type Cx32 and mutant Cx32 forms, the benefits of this approach in *GJB1* mutants, which could result in protein construction, need to be established [64].

#### 3.3.2. Gene Silencing Therapy

RNA interference is technique involving small interfering RNA (siRNA), short hairpin RNA (shRNA), and microRNA (miRNA). RNA interference (RNAi) has been one of the fundamental marks in CMT1A genetic treatment. In vivo, the effectiveness of allele-specific siRNA, which precisely and selectively lessened the level of expression of the mutant allele, was assessed through intraperitoneal administration in Tr-J mice [73]. This treatment significantly improved motor function and muscle volume in the Tr-J mice, as evidenced by results from the rotarod test and magnetic resonance imaging analysis, respectively. When *Pmp22*-targeting siRNA coupled to squalenoyl nanoparticles are intravenously administered in JP18 and JP18/JY13 mice models of CMT1A, they can decrease PMP22 levels and improve locomotor function; nevertheless, the effects are short-lasting and recurrent doses are needed [74]. A long-lasting RNAi therapy was administered by intraneurally injecting an AAV2/9 vector expressing murine *Pmp22*-targeting shRNA in a CMT1A rat model; this model normalized PMP22 and MPZ protein levels and enhanced its function and myelination [75]. The transfection of regular and humanized transgenic neuropathic mouse SCs with a microRNA 29a (miR-29a) expression plasmid decreased the levels of endogenous mouse and transgenic human PMP22 transcripts compared to those in the control vector. Additionally, ectopic expression of miR-29a resulted in a considerable (approximately 50%) decrease in *Pmp22* mRNA, corresponding to an approximate 20% decrease in PMP22 protein levels [76]. Antisense oligonucleotides (ASOs), single-stranded synthetic nucleic acids, specifically bind to mRNA sequences, promoting their degradation in an RNaseH-dependent manner. ASOs have been demonstrated to successfully conquer *Pmp22* mRNA in affected nerves in two murine CMT1A models; notably, starting ASO therapy following the onset of the disease reestablished myelination, motor nerve conduction verocity, and compound muscle action potential nearly to the levels observed in wild-type animals [77].

#### 3.3.3. CRISPR/Cas9

The CRISPR/Cas9 methodology focuses on manipulating regulatory elements in the PMP22 gene to suppress its transcription. This technique was employed in a rat SC line to excise an upstream portion of the Pmp22 gene, a region anticipated to contain an enhancer or promoter for the gene; the elimination of this area resulted in a reduction in Pmp22 mRNA concentrations. The elimination of the TATA-box promoter from the Pmp22 gene in C22 mice via CRISPR/Cas9 caused a downregulation of Pmp22 mRNA and improved the nerve pathology [78].

#### 3.3.4. Hepatocyte Growth Factor: Engenesis^®^ VM202

VM202 represents a nonviral carrier encapsulating a distinct genomic cDNA hybrid derived from the human hepatocyte growth factor (HGF) [79]. HGF facilitates peripheral nerve regeneration through stimulation of SC repair. Specifically, HGF enhances the migration and proliferation of cultured SCs while concurrently upregulating the expression of multiple genes, including glial cell line-derived neurotrophic factor and leukemia inhibitory factor, likely via the activation of the ERK pathways [80]. The FDA has designated VM202 for expedited development and review, otherwise known as the ‘fast track’ status; several clinical trials evaluating this agent for other neurological disorders have been conducted [81,82]. While the outcomes of the CMT1A study (NCT05361031) are yet to be reported, a previous study involving recurrent intramuscular doses of the vector in ischemic heart illness patients observed a reduction in the beneficial effects of VM202 after several months [83], indicating that it may serve as a short-term symptomatic treatment.

## 4. Challenges and Limitations of Clinical Research

Juneja et al. noted a lack of clinical studies and attributed it to several factors. The genetic diversity associated with multiple causative genes and genetic mutations was listed as the primary factor. In addition, the scarcity of cases involving the same gene and symptom heterogeneity failed to stimulate sufficient interest among researchers and pharmaceutical companies. The complex process of transitioning from preclinical studies in animal models to clinical trials in humans was also an obstacle [5]. Moreover, neuropathies characterized by poor prognosis, including amyloid neuropathy, polyneuropathy, visceral hypertrophy, endocrine disorders, M-protein, and cutaneous changes (POEMS) syndrome, offer more quantifiable metrics for their treatment (e.g., survival rates). Conversely, CMT shows a gradual clinical progression, which complicates its prognosis. This condition also exhibits substantial challenges in clinical research and development. Rossor et al. noted that most CMT1A patients had a shared pattern of permanent axonal loss in adulthood, emphasizing the importance of treatments focused on halting or slowing disease progression.

Some clinical outcome metrics for CMT have already been established, including the Charcot–Marie–Tooth Neuropathy Score version 1 (CMTNSv1) adopted in the AA study, and CMTNSv2, which shows a limited ability to identify slight shifts during disease progression. Consequently, alternative metrics (e.g., Rasch-modified CMTNSv2) have been proposed. It is worth noting that the PXT3003 trial employed the Overall Neuropathy Limitation Scale, highlighting the absence of universally accepted measures [84].

AA therapies and their clinical trials evidence the exceptional progress in this field. A significant number of clinical trials based on experimental models are currently exploring a broad spectrum of therapeutic strategies. Although the outcomes of the first extensive multicenter CMT clinical trial were negative, this study provided insights into the necessity for more effective endpoints and biomarker identification. Promising biomarkers such as muscle magnetic resonance imaging and plasma neurofilament light chains may offer valuable information concerning disease activity, burden, and targeted intervention for the most prevalent CMT subtypes.

## 5. Conclusions

Currently, the primary challenge faced by CMT research lies in identifying disease-modifying treatments. To date, no definitive pharmacological treatment for any CMT variant has been established; nonetheless, recent years have witnessed substantial progress. The ongoing investigation of PXT3003, currently the most studied candidate, could yield significant results if it modifies the typically progressive course of the disease and demonstrates a sustained therapeutic effect over an extended treatment period. However, this achievement alone will not suffice. Further advancements from the multitude of ongoing preclinical studies are required to produce symptomatic improvements. Although gene therapy has revolutionized treatment for other neuromuscular diseases, its potential benefits have not been effectively translated to CMT therapy. Hence, future research is anticipated to usher in the much-needed advancement in this area.

## Data Availability

Not applicable.

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
