# Peer review of "The Current State of Charcot–Marie–Tooth Disease Treatment"

_genes, 2023, doi:10.3390/genes14071391_

Round 1
Reviewer 1 Report
This article is a very good and comprehensive review with detailed clinical and genetical classifications of the Charcot-Marie Tooth neuropathy and it also contains a detailed review of the existing molecules and molecules that are currently in clinical studies.
Author Response
Reviewer1:
This article is a very good and comprehensive review with detailed clinical and genetical classifications of the Charcot-Marie Tooth neuropathy and it also contains a detailed review of the existing molecules and molecules that are currently in clinical studies.
→Thank you for your comment. We are pleased to receive your feedback. The reviewers pointed out that some parts of this manuscript showed similarities to another study we submitted to the journal, so it has been edited to reduce such similarities. There are no major differences in content. We would appreciate it if you review our changes.

Reviewer 2 Report
The manuscript of Okamoto and Takashima is an update on currently available and possible therapies for CMT. The introduction gives sufficient background and presents the historic classification of the CMT types and the genetic spectrum of CMT. Further, the authors characterize four main forms of CMT. The results section presents shortly the recent advantages and failures of clinical and preclinical trials. In the summary, the authors look critically at the trials and draw conclusions for the design of the future trials.
Comments:
1. The authors cite/adapted the tables from two good reviews C. Pisciotta et al., 2021 and M.Stavrou et al., Int J Mol Sci. 2021. What is the progress in relation to this review (in the last 2-3 years)? Please state it clearly
2. Please include the limitations of the current trials in a separate paragraph rather than in the conclusions.
3. Please characterize outcome measures, scales etc. used up to date. Are there any developments in this field?
4. What are the main hurdles from genetic therapies? How does cost-effectiveness in terms ofCcMT trials look like? Why are the developments in terms of clinical trials so slow?
Author Response
Reviewer 2:
The manuscript of Okamoto and Takashima is an update on currently available and possible therapies for CMT. The introduction gives sufficient background and presents the historic classification of the CMT types and the genetic spectrum of CMT. Further, the authors characterize four main forms of CMT. The results section presents shortly the recent advantages and failures of clinical and preclinical trials. In the summary, the authors look critically at the trials and draw conclusions for the design of the future trials.
→Thank you very much for your comments. We appreciate your constructive feedback. Below, we have included our opinions on the points you have raised. In addition, the reviewers pointed out that some parts of this manuscript showed similarities to another study we submitted to the journal. Since it is a review paper, we consider it somewhat unavoidable. Nonetheless, we have edited our manuscript to reduce similarities as much as possible. Although there are no major changes in content, we would be happy if you can review the paper again.
Comments:
- The authors cite/adapted the tables from two good reviews C. Pisciotta et al., 2021 and M.Stavrou et al., Int J Mol Sci. 2021. What is the progress in relation to this review (in the last 2-3 years)? Please state it clearly
→Thank you for your comment. As you have mentioned, two excellent reviews on CMT have recently been published, one of which is on the CMT website, and the other is on the CMT website, which is on the CMT website. Although several preclinical studies have already been published, we did not find their citations in these reviews. We deeply analyzed the best parts of these two papers to develop a review focused on the major CMT types. In addition, aspects of curcumin treatment, including those where I was involved have been included in the study.
- Please include the limitations of the current trials in a separate paragraph rather than in the conclusions.
→Thank you for your comment. Accordingly, we have included a new paragraph on the limitations of treatment research. We have included this info in the Challenges and Limitations of Clinical Research section.
- Please characterize outcome measures, scales etc. used up to date. Are there any developments in this field?
→Thank you for your comment. We did not find significant progress regarding factors such as outcome measures and scales in the literature. Nonetheless, we have added this information in the treatment limitations section.
